# Impact of Macroeconomic, Governance and Risk Factors on FDI Intensity—An Empirical Analysis

**K. S. Sujit** [1] , **B. Rajesh Kumar** [2] and **Sarbjit Singh Oberoi** [3,*]

1   Department of Economics, Institute of Management Technology, Dubai 345006, UAE; sujit@imt.ac.ae
2   Department of Finance, Institute of Management Technology, Dubai 345006, UAE; rajesh@imt.ac.ae
3   Department of Operations Management, Institute of Management Technology, Nagpur 440035, India
*   Correspondence: ssingh@imtnag.ac.in

**Abstract:** The study analyzes the impact of macroeconomic, governance and risk factors on foreign direct investment (FDI) intensity with respect to the US market during the period 1960–2019. The study adopted regression methodology. The FDI, macroeconomic and risk data were sourced from the Federal Reserve Economic Data (FRED) database. The governance data were collected from the World Bank Governance Database. The study suggests that infrastructural investments lead to higher FDI. A stronger Euro leads to higher FDI activity in the United States. Research & Development investments is a significant factor which contributes towards enhanced FDI activity. The higher the corporate profitability, the greater the FDI inflows. Exports and imports are significant factors which determine FDI in markets like USA. Inflation has a negative impact on FDI flow regulations, which are aimed to promote private sector development is negatively related to FDI intensity. FDI activity by firms tend to be lower when corruption levels are higher in the country. The higher the governance perception in terms of voice and accountability of citizens, the greater the propensity to attract FDI. The perception of the effectiveness of a government's commitment towards the quality of public and civil services is directly related to FDI investment.

**Keywords:** FDI intensity; macroeconomic factors; risk factors; governance factors

## 1. Introduction

Foreign direct investment (FDI) basically refers to establishment of new firms, acquisitions of companies or assets. Physical investments made directly to the owners of assets in another country are termed as foreign direct investments. Basically, countries strategize incentive policies to attract FDI in their respective countries (Wei and Zhu 2007). FDI investments, send a positive signal regarding the economic prospects and attractiveness of investment in a country. FDI investments are directly related to the pulses of financial markets. FDI is a significant contributor to economic growth of both developed and developing countries (Hiratsuka 2006; Tan et al. 2018; Diana et al. 2019). FDI investments can be analyzed from the viewpoint of cost of capital or investment portfolio theory and industrial organization theory (Lin 1996). According to the cost of capital approach FDI inflow to any country is an evaluated decision based on the criterion of the incremental expected returns vis-a-vis the marginal cost of capital. The industrial organizational theory approach suggests that investment activity of multinational firms is a function of the strategic behaviors which firms adopt in terms of investment activity (Lin 1996). Investment decisions by firms are adopted on the basis of scenario analysis in which firm specific advantages are compared in terms of the costs of investments in diverse locations. Firms adopt strategies of establishment of a foreign subsidiary through FDI to gain technological, managerial, marketing skills, accessibility to markets. In a competitive environment, countries strive for gaining larger share in global productive activity associated with a particular industry through

trade and FDI policies. In terms of flow mechanism, FDI is defined as a movement of capital and other resources from a parent corporation in the home country to the subsidiary company which is created through substantial equity interest in the firm established in the host country (Pugel 1981). FDI flow is largely industry specific and hence the determinants of the industry pattern of FDI is directly related to characteristics of market structure and market conduct across industries (Pugel 1981).

This study examines the determinants of FDI with reference to the US market. The FDI, macroeconomic and risk data were sourced from the FRED database. The governance data were collected from the World Bank Governance Database. The study covered the period 1960–2019. The results were estimated using the ordinary least square regression method. The Breusch–Godfrey serial correlation Lagrange Multiplier (LM) test was conducted for testing serial correlation for the five models. The stability of the model was tested using the cumulative sum (CUSUM) test.

## 1.1. FDI in US

The United States of America (USA) is always in the front place as far as hosting foreign direct investment (FDI) from the world. The major inflow of FDI comes from the United Kingdom, Canada, Japan, Netherlands, Luxembourg, Germany and Switzerland. USA is also ranked very high in terms of outflow of FDI to rest of the world. The reason for attracting huge FDI is due to a very conducive environment of doing business. Eleven indicators that Doing Business-2020 report published by the World Bank Group has evaluated across 190 countries and ranked USA sixth with 84 points.[1] USA has a huge market base with ever increasing demand for goods and services, it has the largest financial market, and is an economic powerhouse of the world with leading manufacturers. The country has all the recipes to attract FDI from across the world.

The policies were transparent and welcoming before turning out to become more protectionist recently. In the later part of the year 2017, Congress also adopted the United States Foreign Investment Review Act of 2017 which allow the Department of Commerce to examine the economic effects of certain foreign investments. Through this, certain sectors were particularly targeted as they are regarded as strategic for the USA. Along with adoption of protectionist policy, Covid-19 has further reduced the flow of FDI to USA. Since the last four years, there has been a decline in inflow of FDI to USA. The World Investment Report 2020 forecasted that the FDI would further shrink by around 20 to 35 percent.[2]

Global FDI flows are forecasted to reduce by 20% in the year 2020 as per World Investment Report, 2020. The pandemic is turning out to be a supply, demand and policy shock for FDI. It is expected to rebound by the year 2022 with a more optimistic and liberalized policies, especially for FDI from USA and the world.

## 1.2. Objective of the Study

The study aims to analyze the determinants of FDI in a major market like US. The study basically examines the impact of macroeconomic, governance and risk factors on FDI during the period 1960–2019.The factors influencing FDI are analyzed after classifying the factors into macroeconomic, governance and risk factors. Identification of factors within this framework is integral for policy analysis.

The factors influencing FDI are also heterogeneous, it is time specific and also country specific. Hence, more country specific studies are needed so as to formulate policies with respect to the specific country.

---

[1]   Doing Business-2020 report, World Bank Group, https://www.doingbusiness.org/content/dam/doingBusiness/country/u/united-states/USA.pdf.

[2]   World Investment Report, 2020, United Nations Conference on Trade and Development, 30th anniversary edition, United Nations. https://unctad.org/system/files/official-document/wir2020_en.pdf.

## 2. Review of Literature

Literature on FDI is large and varied as this has been an engine of growth during the liberalization of economies during the nineties. The researchers have adopted various strategies to capture the impact of FDI and also various factors determining it. For the impact of FDI study, the major research question was does FDI generate growth? (Bermejo Carbonell and Werner 2018). One of the research gaps identified was to have single country analysis due to a heterogeneous relationship between FDI and growth. Especially in the case of developed countries, it is not very clear. Most of the studies have reported a negative relationship; Herzer (2012), an inconclusive relationship; De Mello (1999), and a positive relationship; Jyun-Yi and Chih-Chiang (2008) Olofsdotter (1998). Pelinescu and Radulescu (2009) found the relationship to be indirect with increase in competitiveness and productivity.

Studies have explored the determinants of attracting FDI using a group of countries through panel data (Jyun-Yi and Chih-Chiang 2008; Alfaro 2003; Demirhan and Masca 2008). Most of the studies have analyzed the factors influencing FDI in developed markets (Yeaple 2003) and in developing countries (Demirhan and Masca 2008; Nguyen and Doan 2016). The study by Demirhan and Masca (2008) based on FDI flows to 38 countries suggest that the growth rate of per capita GDP, infrastructural facilities such as telephone connections and degree of openness are factors that positively influences FDI flows. The country specific studies on the factors affecting FDI were limited due to lack of available data in developing countries. However, it has recently gained popularity in developing countries with the availability of time series data—Gheorghe and Vasile (2012); Mohamed Ibrahim Mugableh (2015); Bilgili et al. (2012); Singhania and Gupta (2011); Kyrkilis and Pantelidis (2003); and Kisto (2017).

These country specific studies have uncovered very interesting facts specific to that country for example, Singhania and Gupta (2011) found 63% of FDI flows into India were explained by macroeconomic variables such as GDP, inflation rate, interest rate, patents, money growth, and foreign trade), while the other 37% of FDI flows into India remained unexplained. Kyrkilis and Pantelidis (2003) studied time series of nine countries and found that real gross national product was the most important determinant of outward FDI. Kisto (2017) found that inflation rates and exchange rate are among the major and important factor that determine the inflow of FDI in Mauritius. Bilgili et al. (2012) studied quarterly time series data of Turkey and found that GDP growth, energy prices, exports, imports, country risks, and labor costs significantly influenced FDI flows into Turkey. Grosse and Trevino (1996) studied twelve-year time series of USA and found that home country's exports to the United States and market size of home country positively influence FDI whereas home country's imports from the United States, the cultural and geographic distances of the home country from the United States, and the exchange rate was having negative influences on FDI.

Other important studies focusing on specific country based FDI studies show heterogeneous relationship and varied determinants of FDI. Factors which determine the attractiveness of FDI are different in different countries (Sandhu and Gupta 2016). Gross domestic product per capita of a host country is a determinant of FDI from Singapore to countries such as China and Hong Kong (Leong and Lee 2019). Market seeking variables such as GDP, GDP per capita, and openness to trade are determinants of China's FDI (Nguyen and Doan 2016). Natural seeking FDI is the significant factor for FDI in Japan and Korea while technological acquisition is the relevant factor for FDI in Taiwan (Fung et al. 2009). FDI facilitates the creation of development projects which enhances the level of productivity and employment opportunities in host countries (Travalini 2009; Zvezdanovic 2013). Aliber (1970) advocated the FDI theory suggesting the linkage of the purchasing power of various currencies of the world. His theory suggested that stronger and stable currencies attract FDI inflow to their countries as compared to countries with weaker currencies. The biggest recipients of FDI in Africa are the oil producers (Onyeiwu and Shrestha 2004). A high supply of skilled labor, human capital and low cost of production attracts FDI (Moreira 2009). The study by Lin (1996) suggest that higher Japanese currency wealth and the lower after tax industry profit rate in Japan are determinants of the amount of new Japanese direct investment into US manufacturing industries. The study further suggests that US non-trade barriers induce inward Japanese direct investments to the US. There exist a

positive long run relationship between intra-ASEAN OFDI and its determinants in terms of FDI inflows into the region, host market size of member states, political stability and degree of trade openness of the regional economy (Tan et al. 2018). The study by Cavallari and D'Addona (2013) examines the role of output fluctuations and exchange rate volatility in driving US FDIs. This study finds the evidence of a positive relation between US FDI and host country's cyclical conditions using the sample FDI data of 46 countries over the period 1982–2009. Empirical studies have examined the relation between FDI and exchange rates. The relationship between FDI and exchange rates is highly unstable (Stevens 1998). Exchange rate volatility has a positive impact on FDI (Cushman 1988; Goldberg and Kolstad 1995; Zhang 2003). Pugel (1981) find empirical support for four sources of ownership specific advantages in favor of foreign direct investment, new technology created through research and development, marketing abilities, organizational techniques and capital cost advantages.

Bergstrand and Egger (2006) suggest the importance of a more rigorous and systematic treatment of trade costs in the intra-industry trade literature. The study by Baltagi et al. (2008) suggest that multinational firms' integration strategies are complex and degree of vertical integration varies in a multilateral world. The study by Jang (2011) on the basis of knowledge capital model find support for the hypothesis that a bilateral FTA has negative effects on bilateral FDI in developed–developed country pairs, but positive effects in developed–developing country pairs. Anderson and Sutherland (2015) analyze the impact of investment promotion agencies (IPAs) on attracting emerging market FDI to developed economies by examining Chinese FDI into Canada and find evidence for the fact that the presence of Canadian provincial level IPAs located in China increases the likelihood of Chinese firms locating in that Canadian province.

Keeping all these heterogeneous factors affecting country specific FDI, it is important to study specific countries. The factors affecting FDI in different countries are varying significantly. The present study is an attempt in this direction to study the factors affecting FDI in US economy.

## 3. Data and Methodology

The FDI data, macroeconomic data and risk data was sourced from FRED database. The governance database was taken from the World Bank Governance Database. The period of analysis is 1960–2019. The World Bank's governance data was available from 1996. The dependent variable FDIGDP was regressed upon macroeconomic, governance and risk variables. Yearly data is used for analysis.

### 3.1. General Model

In this model the impact of FDI on different factors representing macroeconomic variables, risk and governance factors are analyzed.

The model is analyzed using the following equation

$$
\begin{aligned}
FDIGDP_t = \alpha_1 \quad & + \beta_1 LTGB10_t + \beta_2 EMPLOY_t + \beta_3 EURDEX_t + \beta_4 INFLA_t + \beta_5 INFRA_t \\
& + \beta_6 POPUGR_t + \beta_7 RGDICFC_t + \beta_8 RDI_t + \beta_9 CPGDI_t \\
& + \beta_{10} NETEXPGDP_t + \beta_{11} NETIMGDP_t + \beta_{12} SMCAPGDP_t + \beta_{13} NFCI_t \\
& + \beta_{14} CBOEVIX_t + \beta_{15} EMVTI_t + \beta_{16} VA_t + \beta_{17} PS_t + \beta_{18} GOVTEFF_t \\
& + \beta_{19} REGQUA_t + \beta_{20} CCORRUP_t + \beta_{21} RULELAW_t + \mu_t
\end{aligned}
$$

where, $\alpha$ *and* $\beta s$ empty are the parameters to be estimated and $\mu_t$ is the error term. The variables used in the model are explained in Table 1.

**Table 1.** Highlights the variables used and its definition.

| Sl. No. | Variable Code | Explanation about Variable | Type of Variable |
|---|---|---|---|
| 1 | **FDIGDP** | FDI net inflows as per cent of GDP. | Dependent |
| 2 | **LTGB10** | 10 year long term government bond yield for US in Percent Annual | Macroeconomic |
| 3 | **EMPLOY** | Employment level, log of thousands of persons self-employed in all industries. | |
| 4 | **EURDEX** | Euro dollar exchange rate | |
| 5 | **INFLA** | Annual inflation based on consumer prices in Percent | |
| 6 | **INFRA** | Infrastructural proxy based on mobile cellular subscriptions in the US, number per 100 people. | |
| 7 | **POPUGR** | Population growth in USA, percent change at annual rate | |
| 8 | **RGDICFC** | Log of consumption value of fixed capital under the category of real gross domestic investment in billions of dollars. | |
| 9 | **RDI** | Log of research and development investment in billion dollars. | |
| 10 | **CPGDI** | Corporate profits as share of gross domestic income, percent annual. | |
| 11 | **NETEXPGDP** | Net exports of goods and services as percent of shares of gross domestic product, annual | |
| 12 | **NETIMGDP** | Net imports of goods and services as percent of shares of gross domestic product, annual | |
| 13 | **SMCAPGDP** | Stock market capitalization to GDP for USA, percent, annual | |
| 14 | **NFCI** | National Financial Condition Index | Financial and Risk |
| 15 | **CBOE VIX** | Chicago Board of Equity Volatility Index | |
| 16 | **EMVTI** | Equity Market Volatility Tracker Index | |
| 17 | **Voice and Accountability (VA)** | Reflects the perceptions of the extent to which a country's citizens are able to participate in selecting their government as well as freedom of expression, freedom of association and free media | Governance Indicator |
| 18 | **PS** | Political stability and absence of violence measures the perception of likelihood of political instability and /or politically motivated violence. | |
| 19 | **GOVTEFF** | Reflects the perception of the quality of public services, the quality of the civil service and the degree of its independence from political pressures, the quality of policy formulation and implementation and the credibility of the government's commitment to such policies. | |
| 20 | **REGQUA** | Reflects the perceptions of the ability of the government to formulate and implement sound policies and regulations that permit and promote private sector development. | |
| 21 | **CCORRUP** | Reflects the perceptions of the extent to which public power is exercised for private gain which includes both petty and grand forms of corruption as well as capture of the state by elites and private interests. | |
| 22 | **RULELAW** | Reflections the perceptions of the extent to which agents have confidence in and abide by the rules of society, and in particular the quality of contract enforcement, property rights, the right police, and the courts, as well as the likelihood of crime and violence. | |
| 23 | **CCORRUP** | Reflects the perceptions of the extent to which public power is exercised for private gain which includes both petty and grand forms of corruption as well as capture of the state by elites and private interests. | |
| 24 | **RULELAW** | Reflections the perceptions of the extent to which agents have confidence in and abide by the rules of society, and in particular the quality of contract enforcement, property rights, the right police, and the courts, as well as the likelihood of crime and violence. | |

The Worldwide Governance Indicators (WGI) are a research dataset which summarizes the views on the quality of governance which are provided by a large number of enterprise, citizen and expert survey respondents in industrial and developing countries. These data are gathered from a number of survey institutes, think tanks, non-governmental organizations, international organizations and private sector firms. These data are collated by the World Bank. WGI project constructs aggregate indicators of six broad dimensions of governance: voice and accountability; political stability and

absence of violence/terrorism; government effectiveness, regulatory quality, rule of law and control of corruption. The six aggregate indicators are based on over 30 underlying data sources reporting the perceptions of governance of a large number of survey respondents and expert assessments worldwide.

*3.2. Analysis and Interpretation*

The analysis is based on yearly data as shown in Table 2. The mean FDI as percent of GDP was 1.19. The mean 10 year long term government bond interest rate was approximately 6 percent during the period 1960–2019. The average inflation during the period was 3.71 per cent and the average population growth rate was 1%. The average corporate profit as percent of national income was 8.17%. The estimates of governance factors ranges from approximately −2.5 (weak) to 2.5 (strong) in terms of governance performance.

**Table 2.** Descriptive statistics.

| | Mean | Median | Standard Deviation | Sample Variance | Kurtosis | Skewness | Range | Minimum | Maximum | Count |
|---|---|---|---|---|---|---|---|---|---|---|
| FDIGDP | 1.19 | 1.10 | 0.85 | 0.72 | −0.05 | 0.67 | 3.34 | 0.07 | 3.41 | 52 |
| LTGB10 | 5.99 | 5.65 | 2.87 | 8.24 | 0.28 | 0.76 | 12.11 | 1.80 | 13.91 | 61 |
| EURDEX | 1.20 | 1.21 | 0.16 | 0.03 | −0.55 | −0.33 | 0.58 | 0.90 | 1.47 | 22 |
| EMPLOY | 3.96 | 3.98 | 0.06 | 0.00 | −0.56 | −0.81 | 0.18 | 3.85 | 4.03 | 61 |
| INFLA | 3.71 | 2.95 | 2.76 | 7.60 | 3.10 | 1.70 | 13.90 | −0.36 | 13.55 | 61 |
| INFRA | 44.52 | 22.75 | 48.05 | 2309.15 | −1.21 | 0.60 | 136.60 | 0.00 | 136.60 | 46 |
| POPUGR | 1.00 | 0.96 | 0.24 | 0.06 | 0.44 | 0.22 | 1.18 | 0.47 | 1.66 | 60 |
| RGDICFC | 2.45 | 2.50 | 0.19 | 0.04 | −1.21 | −0.22 | 0.65 | 2.07 | 2.72 | 61 |
| RDI | 2.09 | 2.22 | 0.52 | 0.27 | −1.20 | −0.36 | 1.73 | 1.10 | 2.83 | 61 |
| CPGDI | 8.17 | 8.20 | 1.45 | 2.09 | −0.67 | 0.20 | 5.90 | 5.50 | 11.40 | 61 |
| NETEXPGDP | −1.70 | −1.30 | 1.82 | 3.31 | −0.82 | −0.33 | 6.60 | −5.60 | 1.00 | 61 |
| NETIMGDP | 10.66 | 10.50 | 4.13 | 17.02 | −1.10 | −0.15 | 13.40 | 4.00 | 17.40 | 61 |
| SMCAPGDP | 126.14 | 130.96 | 16.70 | 279.05 | −0.85 | −0.42 | 60.45 | 92.76 | 153.21 | 25 |
| VA | 1.58 | 1.60 | 0.06 | 0.00 | 0.36 | −1.11 | 0.20 | 1.44 | 1.64 | 25 |
| PS | 0.53 | 0.49 | 0.33 | 0.11 | 0.05 | −0.27 | 1.31 | −0.23 | 1.08 | 25 |
| GOVTEFF | 1.59 | 1.55 | 0.11 | 0.01 | −0.60 | 0.85 | 0.34 | 1.46 | 1.80 | 25 |
| REGQA | 1.52 | 1.57 | 0.15 | 0.02 | −0.93 | −0.34 | 0.50 | 1.26 | 1.76 | 25 |
| CCORRUP | 1.46 | 1.40 | 0.17 | 0.03 | −0.74 | 0.57 | 0.61 | 1.22 | 1.83 | 25 |
| RULELAW | 1.58 | 1.60 | 0.06 | 0.00 | 0.36 | −1.11 | 0.20 | 1.44 | 1.64 | 25 |
| NFCI | 0.00 | −0.33 | 0.91 | 0.84 | 1.95 | 1.64 | 3.60 | −0.97 | 2.63 | 50 |
| CBOE VIX | 19.59 | 17.54 | 6.25 | 39.04 | −0.47 | 0.69 | 21.79 | 11.09 | 32.88 | 31 |
| EMVTI | 19.97 | 18.63 | 5.59 | 31.23 | 0.72 | 0.95 | 25.14 | 10.32 | 35.46 | 36 |

*3.3. Unit Root Test for Stationarity*

A time series is considered to be stationary if all the statistical characteristics of that series are unchanged by shifts in time. In other words, the mean, variance and covariance of a stationary time series does not vary systematically over time. The presence of a unit root shows that the time series is nonstationary. The process of unit root testing is as follows:

$$Y_t = \alpha Y_{t-1} + \epsilon_t$$

a typical time series equation

$$Y_t - Y_{t-1} = \alpha Y_{t-1} - Y_{t-1} + \epsilon_t$$

subtracting $Y_{t-1}$ both the side

$$\Delta Y_t = (\alpha - 1)\Delta Y_{t-1} + \epsilon_t$$

Now by estimating above time series equation, one can test for null hypothesis that $\alpha - 1 = 0$. If this is true, then $\alpha = 1$ which means there is the presence of unit root and the series is nonstationary. In also means that if null hypothesis is rejected ($\alpha < 1$) then the series is stationary. Under the null hypothesis estimated, t value of the coefficient follows the tau ($\tau$) statistics Dickey and Fuller (1979). This test is otherwise known as Dickey Fuller (DF) test and MacKinnon critical values for tau distribution is used for hypothesis testing MacKinnon (1991). The Dickey Fuller test was revised by augmenting the equation by including lagged value of dependent variable which is popularly known as augmented dickey fuller teat (ADF). One of the assumption of DF test is that the error term is

independently and identically distributed but there is a possibility of serial correlation in the error term. This is taken care of by adding lagged difference terms of the dependent variable. A different approach is suggested by Phillips and Perron (PP) to deal with serial correlation in the error term by taking non-parametric statistical method without the lagged difference term (Gujarati 2002). For this study, Dickey Fuller-generalized least square (DF-GLS) test is used for checking the unit root proposed by Elliott et al. (1996). This method is similar to ADF except that the time series is transformed via a generalized least squares (GLS) regression before performing the test. This test has significantly greater power than the previous versions of the augmented Dickey–Fuller test. The result of all three methods are presented in Table 3.

**Table 3.** Unit root test for stationarity.

| Variable | ADF | | PP | | DF-GLS (Max Lag Length 4) | |
|---|---|---|---|---|---|---|
| | With Drift and Trend | | With Drift and Trend | | Tau | Lag Length |
| | I(0) | I(0) with Lag1 | I(0) | I(0) with Lag1 | | |
| FDIGDP | −3.59 ** | −4.22 * | −3.59 ** | −3.78 ** | −4.18 * | 1 |
| LTGB10 | −1.79 | −2.07 | −1.79 | −1.86 | −1.63 | 3 |
| EURDEX | −1.26 | −1.82 | −1.26 | −1.35 | −1.27 | 1 |
| EMPLOY | −1.16 | −1.55 | −1.16 | −1.35 | −1.26 | 1 |
| INFLA | −2.8 | −3.54 ** | −2.8 | −2.98 | −2.99 | 2 |
| INFRA | −2.07 | −1.89 | −2.07 | −1.92 | −1.92 | 4 |
| POPUGR | −2.22 | −2.78 | −2.22 | −2.44 | −2.21 | 1 |
| RGDICFC | −1.23 | −3.66 ** | −1.23 | −1.5 | −3.46 ** | 3 |
| RDI | −0.48 | −1.08 | −0.48 | −0.68 | −0.89 | 3 |
| CPGDI | −2.34 | −3.25 ** | −2.34 | −2.65 | −3.33 ** | 1 |
| NETEXPGDP | −1.72 | −2.14 | −1.72 | −1.96 | −2.26 | 1 |
| NETIMGDP | −2.11 | −1.68 | −2.11 | −2.06 | −2.07 | 1 |
| SMCAPGDP | −2.46 | −3.04 | −2.46 | −2.63 | −3.12 *** | 3 |
| VA | −3.55 ** | −3.32 *** | −3.55 ** | −3.54 ** | −2.71 | 1 |
| PS | −2.02 | −2.34 | −2.02 | −2.13 | −2.45 | 1 |
| GOVTEFF | −3.76 ** | −4.69 * | −3.76 ** | −3.76 ** | −2.85 | 1 |
| REGQA | −2.45 | −2.81 | −2.45 | −2.56 | −2.77 | 1 |
| CCORRUP | −2.24 | −2.68 | −2.24 | −2.31 | −2.28 | 4 |
| RULELAW | −3.55 ** | −3.32 *** | −3.55 ** | −3.54 ** | −2.71 | 1 |
| NFCI | −3.45 ** | −4.17 * | −3.45 ** | −3.67 ** | −4.033 * | 1 |
| CBOE VIX | −2.2 | −2.79 | −2.2 | −2.44 | −2.93 | 1 |
| EMVTI | −2.82 | −2.86 | −2.82 | −2.06 | −2.38 | 1 |

Note: * Significant at 1% level; ** significant at 5% level; *** significant at 10% level.

The result shows that FDIGDP, VA, GOVTEFF, RuleLaw, NFCI are stationary both at l(0) with no lag as well as 1 period lag. RGDICFC, INFLA and CPGDI are stationary at least with one period lag. Rest of the variables are non-stationary in both no lag as well as one period lag. It is interesting to note that with one lag as well the results are mostly the same. As we have mostly non-stationary series, any regression with non-stationary series can give spurious results. To convert these series into stationary, we have used the first difference transformation which will make the series stationary. The test with first difference is presented in Table 4.

The test for unit root result of first difference presented in Table 4 shows that almost all the series is stationary with first difference with drift and trend. The series RGDICFC is not stationary with first difference but it is stationary with second difference. As the results are consistent with both ADF and PP the study has not further proceeded with DF-GLS for the first difference series.

**Table 4.** Unit root test for the variables with first difference.

| Variable | ADF | | PP | |
|---|---|---|---|---|
| | **With Drift and Trend** | | **With Drift and Trend** | |
| | **I(1) with no Lag** | **I(1) with 1 Lag** | **I(1) with no Lag** | **I(1) with 1 Lag** |
| **FDIGDP** | −6.79 * | −5.76 * | −6.79 * | −6.79 * |
| **Ltgb10** | −6.68 * | −6.63 * | −6.68 * | −6.70 * |
| **Eurdex** | −4.31 * | −4.13 * | −4.31 * | −4.33 * |
| **Employ** | −4.85 * | −3.96 ** | −4.85 * | −4.85 * |
| **Infla** | −6.41 * | −7.31 * | −6.41 * | −6.45 * |
| **Infra** | −3.87 ** | −3.33 ** | −3.87 * | −3.90 ** |
| **Popugr** | −5.05 * | −5.04 * | −5.05 * | −5.11 * |
| **Rgdicfc** | −1.7 | −2.98 | −1.7 | −2 |
| **Rdi** | −4.17 * | −3.70 ** | −4.17 * | −4.19 * |
| **Cpgdi** | −5.73 * | 5.99 * | −5.73 * | 5.80 * |
| **Netexpgdp** | −6.68 * | −4.71 * | −6.68 * | −6.68 * |
| **Netimgdp** | −8.77 * | −7 * | −8.77 * | −8.80 * |
| **Smcapgdp** | −3.69 ** | −3.80 ** | −3.69 ** | −3.69 ** |
| **Va** | −5.69 * | −4.32 * | −5.69 * | −5.71 * |
| **Ps** | −4.22 ** | −3.02 | −4.22 ** | −4.22 ** |
| **Govteff** | −6.02 * | −4.74 * | −6.02 * | −6.09 * |
| **Regqa** | −4.41 ** | −3.37 *** | −4.41 ** | −4.41 ** |
| **Ccorrup** | −4.56 * | 3.70 ** | −4.56 * | −4.56 * |
| **Rulelaw** | −5.69 * | −4.23 ** | −5.69 * | −5.71 * |
| **Nfci** | −6.47 * | −6.53 * | −6.47 * | 6.49 * |
| **Cboe vix** | −3.61 ** | −2.89 | −3.61 ** | −3.61 ** |
| **Emvti** | −5.34 * | −4.13 ** | −5.34 * | −5.32 * |

Note: * Significant at 1% level; ** significant at 5% level; *** significant at 10% level.

### 3.4. Unit Root Break in FDIGDP

Both the additive and innovative outlier method has been used to find the break. The additive outlier captures sudden change and the innovative outlier method captures gradual change. The null hypothesis of presence of unit root has been rejected for all the series with first difference indicating that series are stationary. The break has been selected using minimized Dickey-Fuller t-statistics, which is a default option in the software as shown in Figure 1.

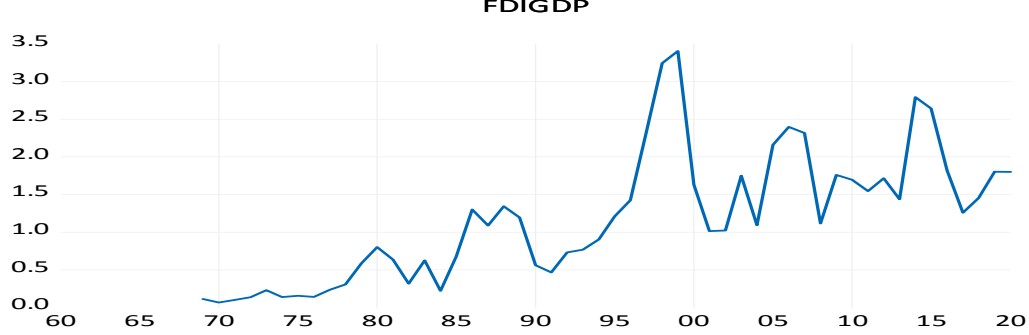

**Figure 1.** Shows the trend with respect to Innovation and Additive Outlier Method.

The result of the unit root with break test shows that there are breaks in the year 1995 to 1997 which shows a significant hike in FDI for USA. The period 1995–1997 has seen as transnational corporations (TNCs) responded to economic growth and continued liberalization in much of the world by further expanding their operations abroad. USA recorded a huge inflow of FDI as well as outflows in these years.

## 4. Regression Results

For the regression analysis, the first difference series is used without any lag as the higher number of independent variables would significantly reduce the degree of freedom. All the variables used in this model are stationary at I(1), i.e., with first difference. Five models have been used out of which the first three models take one type of independent variable. For example, the first model takes all the macroeconomics variables impact on FDI, the second model takes the impact of governance variables and the third model takes financial risk factors impact on FDI. After estimating the result with ordinary least square method, the test for serial correction was conducted using Breusch–Godfrey serial correlation LM test for all the models. The stability of the model was tested using CUSUM test.

The first model takes all the macroeconomic variables as independent variables determining FDI. All the variables are first differenced so that only stationary variables are entered in the model. The result of first model shows that change in infrastructure is highly positive in influencing change in FDI. Any change in log of consumption value of fixed capital under the category of real gross domestic investment in billions of dollars has- a negative impact on changes in FDI. Changes in corporate profits is positively related to changes in FDI. This means that the higher the corporate profits, the more it attracts foreign direct investment. At the same time, exports are also positively related, indicating that export led inward foreign direct investment works well. Higher export opportunities attract foreign direct investment.

Model-2 takes all the governance variables as determinants of foreign direct investment. It is interesting to note that voice and accountability (VA) is positively related which indicates that FDI comes to that country where people, government have any accountability. Perception about the government's ability to formulate sound policy with respect to private sector development has a negative impact on FDI and as expected corruption has a negative impact on FDI.

Model-3 takes all the financial risk factors as determinants of FDI. This model did not find any significant determinant financial and risk variables affecting FDI.

Model-4 started initially with all the macroeconomic and governance variables together and dropped a few variables which were insignificant to check if the results could improve. The final model is presented in Model-4 which clearly shows that most of the variables are showing the same results but some new variables such as imports are now significant in attracting FDI. Inflation is negatively affecting FDI whereas the exchange rate is positively related. The higher importing country provides opportunity for the firms to produce at home and for that FDI helps in getting resources. Yet another important result shows that quality of public services, the quality of the civil service and the degree of its independence from political pressures, the quality of policy formulation and implementation and the credibility of the government's commitment to such policies etc. attracts FDI. Model-5 includes a financial risk model to check if it is important in attracting FDI. The result has not shown any evidence of financial risk affecting FDI as shown in Table 5.

The CUSUM test for stability of each model is also checked the graphical representation shows that all the models are within the critical rage of 5% significance.

As the first difference models are estimated, it is possible that there is a presence of serial correlations in the error term. In order to check serial correlation, the model used Breusch–Godfrey The serial correlation LM test was used to find out the presence of serial correlation. The test shows that in Models 1, 4 and 5 there is serial correlation. For other models (2 and 3) there is no serial correlation. This serial correlation needed to be corrected as serial correlation could provide an inefficient estimator and there is a possibility that the regression coefficient appears to be statistically significant when it is actually not. In order to correct the serial correlation, the model incorporated lagged dependent variables till the result of Breusch–Godfrey serial correlation LM test shows the absence of serial correlation. The result of Model-6 with four lagged dependent variables was used to check serial correlation using the Breusch–Godfrey serial correlation LM Test. The result shows that there is an absence of serial correlation as the observed R-square value is 3.5 and this is not significant. In Model-7 the same process is repeated and with dependent variable with 1 lag turned out to be serial correlation

free as shown by the test value, i.e., 5.21. All the models estimated are stable as tested with CUSUM Test in Table 6.

**Table 5.** First difference models estimated.

| Variables | Model-1 (Macroeconomic Variables) | Model-2 (Governance) | Model-3 (Financial Risk Factor) | Model-4 | Model-5 |
|---|---|---|---|---|---|
| C | −0.76 (−1.38) | −0.09 (−0.67) | 0.04 (0.36) | −0.44 (−1.81) | −0.57 (−1.79) |
| D(LTGB10) | −0.35 (−1.21) | | | | |
| D(EURDEX) | 2.10 (1.21) | | | 2.65 (2.94) ** | 2.27 (2.46) ** |
| D(EMPLOY) | −13.74 (−0.61) | | | | |
| D(INFLA) | −0.06 (−0.26) | | | −0.33 (−2.34) ** | −0.17 (−0.97) |
| D(INFRA) | 0.14 (2.31) ** | | | 0.10 (3.72) * | 0.11 (3.84) * |
| D(RGDICFC) | −107.03 (−1.91) *** | | | −111.54 (−3.93) * | −101.04 (−3.43) ** |
| D(POPUGR) | 0.50 (0.17) | | | | |
| D(RDI) | 36.52 (1.79) | | | 33.24 (3.29) * | 33.55 (2.48) ** |
| D(CPGDI) | 0.87 (3.55) * | | | 0.72 (5.76) * | 0.74 (4.71) * |
| D(NETEXPGDP) | 1.32 (2.05) *** | | | 1.19 (4.01) * | 1.20 (3.73) * |
| D(NETIMGDP) | 0.40 (0.97) | | | 0.59 (2.96) ** | 0.47 (2.25) *** |
| D(SMCAPGDP) | −0.01 (−0.57) | | | −0.01 (−1.17) | −0.01 (−1.11) |
| D(VA) | | 5.54 (2.53) ** | | 4.34 (3.10) ** | 4.32 (3.15) ** |
| D(PS) | | −0.76 (−1.18) | | | |
| D(GOVTEFF) | | 0.97 (0.52) | | 2.29 (2.02) *** | −2.13 (−1.69) |
| D(REGQA) | | −2.76 (−1.90) *** | | | −1.23 (−1.26) |
| D(CCORRUP) | | −3.13 (−1.79) *** | | | 0.29 (0.23) |
| D(NFCI) | | | −0.68 (−1.45) | | |
| D(CBOE_VIX) | | | 0.06 (1.29) | | |
| D(EMVTI) | | | −0.05 (−1.10) | | |
| R-Squared | 0.74905 | 0.395944 | 0.117643 | 0.928631 | 0.947091 |
| Adjusted R-Squared | 0.372625 | 0.228151 | 0.015832 | 0.841402 | 0.848831 |
| Breusch–Godfrey Serial Correlation LM Test: Observed R-Square | 12.83 * | 2.79 | 1.48 | 9.63 * | 7.49 ** |
| Cusum Test for Stability |  |  |  |  |  |

Note: * Significant at 1% level; ** significant at 5% level; *** significant at 10% level.

**Table 6.** Estimation with correction for serial correlation.

| Variables | Model-6 | Model-7 |
|---|---|---|
| C | −0.52 (−2.07) *** | −0.56 (−2.15) *** |
| D(FDIGDP(−1)) | −0.29 (−1.83) | −0.10 (−0.89) |
| D(FDIGDP(−2)) | −0.22 (−1.38) | |
| D(FDIGDP(−3)) | −0.32 (−2.24) | |
| D(FDIGDP(−4)) | −0.11 (−1.05) | |
| D(EURDEX) | 0.90 (0.79) | 2.37 (2.41) ** |
| D(INFLA) | −0.37 (−2.32) *** | −0.20 (−1.08) |
| D(INFRA) | 0.10 (3.81) ** | 0.11 (3.72) * |
| D(RGDICFC) | −101.34 (−3.47) ** | −114.19 (−3.77) * |

**Table 6.** *Cont.*

| Variables | Model-6 | Model-7 |
|---|---|---|
| D(RDI) | 32.41 (3.34) ** | 38.19 (3.90) * |
| D(CPGDI) | 0.55 (3.63) ** | 0.80 (6.75) * |
| D(NETEXPGDP) | 1.26 (4.35) * | 1.23 (3.94) * |
| D(NETIMGDP) | 0.69 (3.31) ** | 0.51 (2.20) *** |
| D(SMCAPGDP) | −0.01 (−1.06) | −0.01 (−1.90) *** |
| D(VA) | 5.45 (3.62) ** | 5.68 (4.60) * |
| D(GOVTEFF) | −0.93 (−0.72) | |
| D(REGQA) | | −1.67 (−1.81) |
| R−squared | 0.9654 | 0.92979 |
| Adjusted R−squared | 0.8616 | 0.824476 |
| Breusch−Godfrey Serial Correlation LM Test: Observed R−square | 3.5 | 5.21 |

Note: * Significant at 1% level; ** significant at 5% level; *** significant at 10% level.

For both the models i.e., Table 6 and Figure 2, a stability test which is popularly known as CUSUM test has been performed. The result of CUSUM test is graphically presented in Figure 2 which shows that it is stable as the line is in the band of 5% significance. As one can see in Figure 2, the CUSUM series is within the upper and lower critical line after the 20th recursive regression, which indicates model stability as shown in Figure 2.

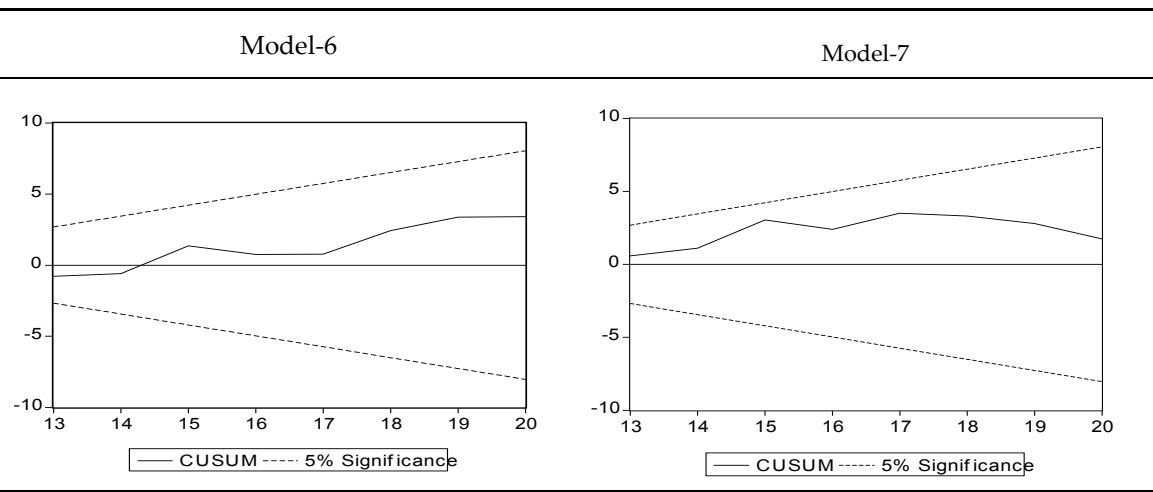

**Figure 2.** CUSUM test for stability of the model.

## 5. Conclusions

This research study analyzes the impact of macroeconomic factors, governance factors and risk factors on FDI activity with respect to the US market. The period of study is 1960–2019. The data are analyzed with respect to annual data for the variables during the period. FDI activity is proxied by the proportion of FDI in terms of GDP. Altogether, eleven factors were included under macroeconomic factors. Three factors were incorporated under risk factors. Six worldwide governance factors in terms of indexes for countries were included under governance factors.

The study suggests that infrastructural investments lead to higher FDI. Higher technological infrastructure in terms of telecommunication facilities promotes FDI activity. A stronger Euro leads to higher FDI activity in the United States. R&D investments are a significant factor which contributes towards enhanced FDI activity. The higher the corporate profitability, the greater the FDI inflows. It can be interpreted that higher profitability of firms sends positive signal which leads to the creation of market that attracts FDI in larger magnitude. Exports and imports are significant factors which determine FDI in markets like USA. Inflation has a negative impact on FDI flows.

Regulations, which are aimed to promote private sector development are negatively related to FDI intensity. FDI activity by firms tend to be lower when corruption levels are higher in the country. The higher the governance perception in terms of voice and accountability of citizens towards governance, the greater the propensity to attract FDI. The perception of the effectiveness of the government's commitment towards the quality of public and civil services is directly related to the FDI investment.

The study also analyzes the impact of risk variables on FDI. The risk variables included in the analysis were proxies representing the financial condition, volatility of the market in terms of derivatives and overall equity market volatility. The results show no significant impact of risk variables on FDI flows.

Our findings with respect to impact of infrastructural investments on FDI is similar to the finding by Demirhan and Masca (2008). The study documents similar results with respect to the impact of foreign trade on FDI as found in the study by Singhania and Gupta (2011). Our results with respect to profitability is similar to the conclusion drawn by Lin (1996). The linkage of exchange rates and FDI was documented by studies such as Zhang (2003); Cavallari and D'Addona (2013). The positive impact of research and development on FDI is also documented by studies such as Pugel (1981).

## 6. Limitations

The study has used annual data. The use of monthly data would have resulted in the availability of more data points for analysis. The study focused on the determinants of inbound FDI. The impact of outbound FDI has not been explored in this paper.

**Author Contributions:** The Conceptualization of the paper was done by B.R.K. and S.S.O. K.S.S. worked on the methodology; the validation of the model and analysis using software was done by all the three authors. The final observations and validation of results has been done by K.S.S. The supervision of the project was done by B.R.K. The final editing and proof reading was done by B.R.K. and S.S.O. All authors have read and agreed to the published version of the manuscript.

**Funding:** This research received no external funding.

**Conflicts of Interest:** The authors declare no conflict of interest.

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
