# Peer review of "Impact of Macroeconomic, Governance and Risk Factors on FDI Intensity—An Empirical Analysis"

_jrfm, doi:10.3390/jrfm13120304_

Round 1

Reviewer 1 Report

The manuscript has improved significantly. 

Reviewer 2 Report

The paper addresses an important and significant issue of the factors that determine the FDI activity, using the example of the US market. The paper's overall quality is high - it is based on the sound methodology and the results were interpreted correctly. There are, though, some minor issues that should be revised:

  1. Some details about the research method and sample should be provided in the abstract and introduction.
  2. Introduction presents some important details but with no references provided.
  3. Conclusion should include comparison to the results of the previous studies and overview of the limitations of the analysis.
  4. The references were not listed alphabetically.
  5. Some parts of the paper require copy-editing.

Author Response

Responses to Reviewer Comments:

All changes highlighted in red in the paper

  1. Some details about the research method and sample should be provided in the abstract and introduction.

Response

Details about research method and sample provided in both abstract and introduction. Lines 10-12;

Line 46-51.

  1. Introduction presents some important details but with no references provided.

Response

References provided: Line 26; line 29-30; line 31, 35, line 43,44.

  1. Conclusion should include comparison to the results of the previous studies and overview of the limitations of the analysis.

Response

Comparison with previous results done. Limitations discussed, line 326-332;333-336.

  1. The references were not listed alphabetically.

References are now alphabetically listed .

  1. Some parts of the paper require copy-editing.

Copy editing done

This manuscript is a resubmission of an earlier submission. The following is a list of the peer review reports and author responses from that submission.

Round 1

Reviewer 1 Report

See attached file

Reviewer 2 Report

I am sorry since my report is a bit rough. In my view, you cannot call this document a research paper. It is really hard to read, without connections between paragraphs, even sentences. There is no motivation. There is no contribution to the literature. There are no reasons supporting the proposed model, which shows up suddenly in the text. The literature review is weak and inconsistent. There are quite a few, if not gross mistakes, confusing sentences regarding the methodology and the results. The abstract does not fulfil the requirements of a good abstract.

In conclusion, there is no way of recommending the publication of this paper. Before sending it to another journal, in my modest opinion, the authors should strive for success.